# When Non-Commutativity Breeds Unfairness: A Geometric–Algebraic View of Uncertainty in VAEs

## Abstract

We propose a novel theoretical framework that unifies geometric and algebraic perspectives on uncertainty quantification in deep generative models. Standard variational autoencoders (VAEs) often underestimate uncertainty in the presence of non-commutative transformations, leading to miscalibrated confidence and potential fairness violations. Our approach introduces an integrated diagnostic and regularization framework that monitors transformation relationships via Baker–Campbell–Hausdorff deviation and decoder order-swap tests; then, we apply category-specific regularization. Commutative pairs receive standard penalties, decoder-induced artifacts are suppressed through equivariance constraints, and genuinely non-commutative transformations are governed by a deformation-stability principle linking commutator strength to required uncertainty scaling. The key theoretical result establishes that geometric uncertainty, measured through the Riemannian sensitivity of the decoder to Lie-algebra actions, should scale with algebraic non-commutativity to ensure proper calibration. This framework closes a gap in generative modelling by providing principled diagnostics and regularization strategies for geometrically-aware uncertainty in symmetry-rich latent spaces, with direct implications for fairness when biased correlations induce spurious non-commutativity.

## 1 Introduction and related work

Uncertainty quantification is essential for trustworthy machine learning, particularly for generative models. Variational Autoencoders (VAEs) are powerful probabilistic models that map complex input data into a lower-dimensional latent space while regularizing these latents to follow a prior distribution. In their basic formulation, latent variables are assumed to be independent or only linearly correlated, yet natural images display strong nonlinear dependencies between latents [5]. While VAEs are valued for their ability to learn rich representations and provide approximate uncertainty estimates via their probabilistic framework, these estimates are often miscalibrated [4]. In particular, standard VAEs can fail to reflect the true geometry of the latent space [7] or the informativeness of the input data [4]. To address these issues, recent work has developed a geometric perspective on VAEs, showing that the latent space can be endowed with a Riemannian structure, where the variational posterior covariance or related information metrics induce a local geometry [1, 6, 7, 18, 21]. Such formulations enrich representation learning by improving interpolation, sampling, and prior design. However, because these Riemannian metrics are derived from symmetric covariance or information matrices, they capture curvature and sensitivity but cannot represent order-dependent effects. Thus, they underestimate uncertainty in settings governed by non-commutative transformations.

A complementary line of work explores symmetry-aware latent representations in VAEs, where transformations are tied to Lie-group structure. Early models such as Lie-Group Sparse Coding (LSC) [8] and the Commutative Lie Group VAE (CLG-VAE) [22] assume commutativity, enabling

tractable inference but excluding non-Abelian settings. Quessard et al. [17] extend to orthogonal transformations in $\mathrm{SO}(n)$, capturing limited non-commutativity under linear actions. Other symmetry-based approaches, including GroupVAE [16], Linear Symmetry-Based Disentanglement (LSBD) [20], and Composite Factor-Aligned Symmetry Learning (CFASL) [13], employ linear or alignment-based constraints but still neglect nonlinear, fully non-commutative group dynamics.

Collectively, these efforts underscore that while both geometric and group-theoretic perspectives enrich latent representation learning, existing frameworks either exclude or only partially capture non-commutative and nonlinear group actions. Thus, the current perspective remains incomplete for spaces equipped with a non-commutative Lie group structure. This oversight has critical consequences for model reliability: even properly calibrated systems with correctly estimated uncertainty are affected by non-commutative transformation structure. Inherent uncertainty increases with commutator strength, causing standard calibration to systematically underestimate it. Discrepancy analysis in our paper further reveals that in the non-commutative case, uncertainty grows faster than can be explained by the accumulated uncertainty of the partial actions. Such underestimation can lead to overconfident predictions in data regions with strong non-commutative effects, potentially resulting in unfair outcomes. Fairness violations can therefore arise when biased correlations induce spurious non-commutativity. This creates geometric complexity that can systematically disadvantages certain subgroups, mirroring recent work that finds a strong link between geometric complexity and model bias/fairness [14]. Thus, this paper makes the following contributions: a) we extend the geometric perspective of VAEs to general Lie groups by introducing a deformation–stability principle. b) this principle establishes a lower bound on reconstruction uncertainty in terms of the commutator norm of the Lie algebra generators, analogous to the Heisenberg uncertainty principle, where non-commuting operators impose a lower bound on measurement precision. c) our framework yields a pull-back metric on the Lie algebra that quantifies the decoder's sensitivity to non-commutative transformations, providing an uncertainty that integrates both algebraic and geometric structure, with direct implications for applications such as algorithmic fairness.

## 2 Diagnostic Framework

The related work reveals a gap in linking algebraic non-commutativity to geometric uncertainty in deep generative models. Here, by geometric uncertainty, we mean uncertainty quantified through the Riemannian geometry of the latent space, induced by the decoder Jacobian. We adapt the methodology from [22] and introduce an adaptive regularization framework that diagnoses latent generator relationships and applies targeted penalties accordingly. Figure 1, provides a modified version of this model which handles both commutative (abelian) and non-commutative group pairs.

### 2.1 Unconstrained training and analysis

We train a Commutative Lie Group VAE (CLG-VAE) [22] without Hessian or commutator penalties. Encoding $x \in \mathcal{X}$ gives $\hat{z} = E_{\mathrm{img}}(x)$ and variational posterior $(\mu, \log \sigma^2) = E_{\mathrm{group}}(\hat{z})$. We sample

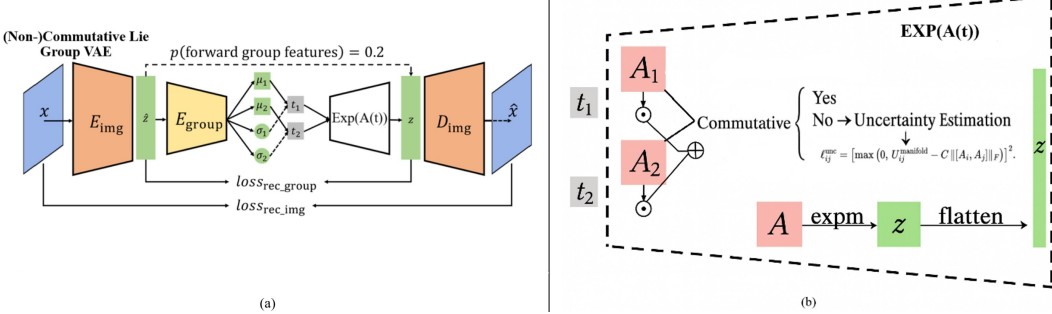

Figure 1: **(Non-)Commutative Lie Group VAE.** (a) Overview of the proposed model extending the Commutative Lie Group VAE (CLG-VAE) to handle both commutative and non-commutative subgroup pairs. An input image is encoded by $E_{\mathrm{img}}$ into a group representation $\hat{z}$, which is passed through the group encoder $E_{\mathrm{group}}$ to obtain Lie algebra coordinates $t$ using the reparameterization trick. These coordinates are mapped through the exponential layer to generate the group representation $z$, then decoded by $D_{\mathrm{img}}$ for reconstruction. (b) The exponential mapping layer, where each scalar $t_i$ is associated with a learnable Lie algebra generator $A_i$. For non-commutative subgroup pairs, the model estimates uncertainty to capture variability in their ordered composition.

74  $t = \mu + \sigma \odot \varepsilon$ with $\varepsilon \sim \mathcal{N}(0, I)$, construct the Lie-algebra element $A(t) = \sum_j t_j A_j$, and obtain the
75  transformed latent $z = \exp(A(t))\hat{z}$, which is then passed to the decoder $D_{\mathrm{img}}$.

76  The core training objective is the unconstrained ELBO:

$$\mathcal{L}_{\mathrm{VAE}}(x) = \ell_{\mathrm{recon}}(D_{\mathrm{img}}(z), x) + \alpha \|z - \hat{z}\|_2^2 + \beta \mathrm{KL}(q(t \mid \hat{z}) \| \mathcal{N}(0, I)).$$

77  Here, $\ell_{\mathrm{recon}}$ is the image reconstruction loss, $\|z - \hat{z}\|_2^2$ is a group reconstruction term enforcing
78  consistency between the encoder output and the Lie-transformed latent representation. $\beta$ scales the
79  KL regularizer controlling the trade-off between reconstruction fidelity and latent regularization.
80  Notation and detailed definitions of variables are provided in Appendix A.1.

## 2.2  Discrepancy analysis: Background

82  The Baker–Campbell–Hausdorff (BCH) formula [12] quantifies the deviation between $e^{A+B}$ and
83  $e^A e^B$, with the leading term involving the commutator $[A, B]$. When $[A, B] = 0$, matrices commute
84  and $e^{A+B} = e^A e^B$; otherwise, non-commutativity introduces higher-order nonlinear BCH terms.
85  Commutator magnitude can be assessed via matrix norms: Frobenius norms capture aggregate
86  effects (used in this study), while spectral or Schatten norms emphasize worst-case deviations [2, 11].
87  Beyond parameter-space measures, Godavarti [10] formalizes non-commutative operators in monoidal
88  categories, explicitly capturing order sensitivity ($A_i A_j \neq A_j A_i$). This framework generalizes order-
89  sensitive architectures such as transformers and structured state-space models, motivating swap tests
90  that quantify non-commutativity by permuting operator order and measuring output changes.

## 2.3  Two-level non-commutativity diagnosis

92  Inspired by recent literature from the previous section, we propose a two-level diagnostic for generator
93  pairs $(A_i, A_j)$ to distinguish group-induced from decoder-induced order effects.

94  **(1) Algebraic test (BCH).** The BCH deviation

$$D_{i,j} \;=\; \left\| \exp(t_i A_i + t_j A_j) - \exp(t_i A_i) \exp(t_j A_j) \right\|_F. \tag{1}$$

95  measures intrinsic algebraic non-commutativity. $D_{ij} \approx 0$ indicates commuting generators.

96  **(2) Decoder test (Order swap).**  We compare reconstructions when applying the generators in
97  opposite orders:

$$\Delta_{ij} \;=\; \mathbb{E}_{\hat{z} = E_{\mathrm{img}}(x),\, t \sim q(t \mid \hat{z})} \left\| D_{\mathrm{img}}\big(\exp(A_i)\exp(A_j)\hat{z}\big) - D_{\mathrm{img}}\big(\exp(A_j)\exp(A_i)\hat{z}\big) \right\|_2. \tag{2}$$

98  Here, $\Delta_{ij} \approx 0$ means the decoder is order-invariant, while large values indicate decoder-induced
99  sensitivity. Combining $D_{ij}$ and $\Delta_{ij}$, we categorize generator pairs as follows:

100  **Commutative**: $D_{ij} \approx 0$, $\Delta_{ij} \approx 0$, both algebra and decoder order-invariant.

101  **Decoder-Induced**: $D_{ij} \approx 0$, $\Delta_{ij} > 0$, abelian, decoder introduces spurious order effects.

102  **Strong**: $D_{ij} > 0$, $\Delta_{ij} > 0$, order effects expected; analysis of decoder alignment with Lie bracket.

103  **True non-Commutative (algebraic)**: $D_{ij} > 0$, $\Delta_{ij} \approx 0$, order sensitivity reflects genuine Lie
104  algebra structure.

105  In practice, we distinguish "$\approx 0$" from "$> 0$" by normalizing $D_{ij}$ and $\Delta_{ij}$ and setting thresholds as
106  high percentiles of their validation distribution (e.g. $90^{\mathrm{th}}$).

107  **Local Jacobian commutator test.** To distinguish whether a strong $\Delta_{ij}$ arises from the group
108  structure or the decoder, we compute a local Jacobian commutator. This probes the decoder's
109  first-order response to generator actions at a point on the data manifold. Details can be found in A.4

## 3  Targeted regularization

111  Here we present category-specific regularization terms for generator pair types from section 2.3 and
112  develop a novel manifold-based uncertainty regularizer grounded in deformation-stability theory.

## 3.1 Regularization terms

Based on the pairwise diagnostics, we apply targeted regularization:

**Commutative** ($D_{ij} \approx 0$, $\Delta_{ij} \approx 0$): Following CLG-VAE, we apply Hessian and commutator penalties to stabilize training and enforce order-consistent actions in latent space

$$\mathcal{L}_{\text{Hess}} = \lambda_{\text{Hess}} \sum_{(i,j)\in\mathcal{I}} \|A_i A_j\|_F^2, \quad \mathcal{L}_{\text{comm}} = \lambda_{\text{comm}} \sum_{(i,j)\in\mathcal{I}} \|[A_i, A_j]\|_F^2.$$

**Decoder-Induced** ($D_{ij} \approx 0$, $\Delta_{ij} > 0$): Algebra is commutative but decoder introduces spurious order sensitivity. We therefore enforce decoder invariance by adapting the consistency loss of [13]:

$$\mathcal{L}_{de}(i,j) = \mathbb{E}_{\hat{z}} \|D_{\text{img}}(\exp(A_i)\exp(A_j)\hat{z}) - D_{\text{img}}(\exp(A_j)\exp(A_i)\hat{z})\|_2^2.$$

**True non-commutative (Algebraic).** ($D_{ij} > 0$, $\Delta_{ij} \approx 0$): We preserve the genuine group structure non-commutativity and apply uncertainty regularizer $\mathcal{L}_{\text{unc}}(i,j)$ to ensure decoder faithfully models geometric variability.

**Strong** ($D_{ij} > 0$, $\Delta_{ij} > 0$): Both algebra and decoder contribute. We separate effects:

- **Algebraic component** (predicted by $[A_i, A_j]$): Treated as genuine structure, contributes to $\mathcal{L}_{\text{unc}}(i,j)$, introduced later under the deformation-stability principle (see Section 3.2.2 for formulation).
- **Decoder-induced residual**: Penalized via alignment loss to enforce the decoder's local commutator $C_{ij}(z)$ to follow the algebraic direction $v_{[ij]}(z)$

$$\mathcal{L}_{\text{align}}(i,j) = \mathbb{E}_{\hat{z},t}[1 - \cos\angle(C_{ij}(z), v_{[ij]}(z))].$$

Thus, uncertainty arises only from true algebra while suppressing spurious decoder effects.

## 3.2 Manifold-based uncertainty regularizer with deformation stability

We link reconstruction uncertainty to generator non-commutativity through deformation-stability theory [3], measuring sensitivity to infinitesimal moves along generator directions via Lie-group actions on latent vectors.

### 3.2.1 Theoretical foundation: From torsion to geometric uncertainty

We build on two key theoretical pillars:

- **Torsion as non-commutativity**: Our approach is aligned with the geometric link between non-commutativity and torsion described by Munthe-Kaas et al. [15], where the torsion tensor $T$ of a connection is defined by the Lie bracket $[X, Y]$. While our model does not instantiate their full post-Lie algebroid structure, the analogy provides a useful correspondence:
  - Our generators $\{A_i\}$ correspond to elements of a Lie algebra $\mathfrak{g}$
  - Our decoder Jacobians $J_k(z)$ play the role of a connection $\nabla$
  - Our commutators $[A_i, A_j]$ act as torsion-like quantities $T$, where $\|[A_i, A_j]\|_F$ quantifies the intrinsic deformation capacity of the generator pair.
  This motivates interpreting commutator norms as geometric torsion, providing an algebraic basis for quantifying non-commutative effects.
- **Deformation-stability principle**: Following Bronstein et al. [3], we require that reconstruction changes be bounded by deformation magnitude. The core insight from deformation-stability states that for a transformation $\tau$, the output change should satisfy:

$$\|f(\rho(\tau)z) - f(z)\| \leq Cc(\tau)\|z\|$$

  where $c(\tau)$ quantifies deformation magnitude. In our Lie-group context: non-commuting generators $A_i, A_j$ create *deformations* of the group action; the torsion-like commutator norm $\|[A_i, A_j]\|_F$ serves as $c(\tau)$; worst-case joint reconstruction sensitivity $U_{ij}^{\text{manifold}}$ replaces $\sigma_i\sigma_j$, which assumes independent sensitivities and underestimates uncertainty for entangled pairs.

**Theorem 1** (Geometric interpretation of reconstruction sensitivity). *The joint sensitivity measure* $U_{ij}^{\text{manifold}} = \sigma_{\max}\left(\left[\frac{\partial x_{\text{rec}}}{\partial t_i} \middle| \frac{\partial x_{\text{rec}}}{\partial t_j}\right]\right)$ *quantifies the worst-case sensitivity of reconstructions to changes in non-commuting generator parameters* $(t_i, t_j)$, *measured through the Riemannian geometry induced by the decoder. Proof for this can be found in A.2*

**Stability postulate.** Theorem 1 provides the geometric foundation for our deformation-stability principle. The $U_{ij}^{\mathrm{manifold}}$ captures the maximum Riemannian distortion induced by non-commuting generators.

To clarify the geometric interpretation, note that $x_{\mathrm{rec}} = D_{\mathrm{img}}(z)$ with $z(t) = \exp(A(t))\,\hat{z}$ being the group-transformed latent code. By the chain rule, $\frac{\partial x_{\mathrm{rec}}}{\partial t_i} = \frac{\partial D_{\mathrm{img}}}{\partial z} \cdot \frac{\partial z}{\partial t_i}$, so our sensitivity measure incorporates both the decoder Jacobian (Riemannian geometry) and the group-action derivative (algebraic structure). Following the analogy between commutators and torsion [15] and the deformation–stability principle [3], we postulate that the worst-case geometric sensitivity is bounded below by torsion-like non-commutativity:

$$\underbrace{U_{ij}^{\mathrm{manifold}}}_{\text{maximum Riemannian distortion}} \geq C \cdot \underbrace{\|[A_i, A_j]\|_F}_{\text{torsion-like non-commutativity}}. \tag{3}$$

where $C > 0$ is a global stability constant. Unlike Theorem 1, this inequality is not a derived result but a modelling assumption that enforces consistency between the algebraic structure of the latent generators and the geometric uncertainty in reconstruction space. We compute $U_{ij}^{\mathrm{manifold}}$ efficiently via Jacobian–vector products and spectral analysis of a $2 \times 2$ Gram matrix (see Appendix A.3 for details).

### 3.2.2 Deformation Hinge Loss and Stabilizing C

We enforce the stability principle via a hinge loss that penalizes insufficient uncertainty for non-commuting generator pairs $(i, j)$:

$$\ell_{ij}^{\mathrm{unc}} = \left[\max\left(0, U_{ij}^{\mathrm{manifold}} - C\,\|[A_i, A_j]\|_F\right)\right]^2.$$

To stabilize the hyperparameter $C$, we set $C = \frac{1}{2}\|A_i\|^2\|A_j\|^2 L$, where $L$ is the decoder's Lipschitz constant. $L$ is bounded by applying spectral normalization to the decoder weights [19], which stabilizes the regularization. To prevent over-regularization, we optionally weight the loss by the Centered Kernel Alignment (CKA) between the generators' output effects [13]. The final loss enforces the geometrically-principled link as in equation 3 and is calculated as:

$$\mathcal{L}_{\mathrm{unc}} = \lambda_{\mathrm{unc}} \sum_{(i,j) \in \mathcal{E}_{\mathrm{non\text{-}commutative}}} w_{ij}\,\ell_{ij}^{\mathrm{unc}}, \quad w_{ij} \in \{1, \mathrm{CKA}_{ij}\}.$$

Let $\mathcal{P}_{\mathrm{comm}}, \mathcal{P}_{\mathrm{dec}}, \mathcal{P}_{\mathrm{alg}}, \mathcal{P}_{\mathrm{strong}}$ be the pair sets from diagnostics. So, the final loss becomes:

$$\mathcal{L}_{\mathrm{total}} = \mathcal{L}_{\mathrm{VAE}} + \sum_{\mathcal{P}_{\mathrm{comm}}} \left[\lambda_{\mathrm{Hess}}\|A_i A_j\|_F^2 + \lambda_{\mathrm{comm}}\|[A_i, A_j]\|_F^2\right]$$

$$+ \sum_{\mathcal{P}_{\mathrm{dec}}} \lambda_{de}\mathcal{L}_{de}(i, j) + \sum_{\mathcal{P}_{\mathrm{alg}}} \lambda_{\mathrm{unc}} w_{ij} \ell_{ij}^{\mathrm{unc}}$$

$$+ \sum_{\mathcal{P}_{\mathrm{strong}}} \left[\lambda_{\mathrm{unc}} w_{ij} \ell_{ij}^{\mathrm{unc}} + \lambda_{\mathrm{align}}\mathcal{L}_{\mathrm{align}}(i, j)\right].$$

where each pair contributes only its category-specific terms.

## 4 Conclusion

We establish a unified geometric-algebraic framework showing that non-commutative structures in VAE latent spaces impact uncertainty quantification. Our deformation-stability principle demonstrates that geometric uncertainty should scale with algebraic non-commutativity. This addresses systematic uncertainty underestimation that might become fairness-critical when biased correlations induce spurious non-commutativity. Further discussion of fairness emerging from geometric complexity is provided in Appendix A.5. Future work will validate our framework using: (1) commutator-strength correlation to verify that geometric uncertainty scales with algebraic non-commutativity across generator pairs, and (2) deformation-stability compliance rates to test whether the theoretical bound $U_{ij}^{\mathrm{manifold}} \geq C \cdot \|[A_i, A_j]\|_F$ holds empirically. Additionally, we will evaluate whether geometric uncertainty $U_{ij}^{\mathrm{manifold}}$ predicts reconstruction degradation by measuring its correlation with reconstruction error in regions with strong non-commutativity, and ensure sample quality remains competitive via Fréchet Inception Distance (FID) benchmarks.

# A   Technical Appendices and Supplementary Material

## A.1   Definitions of Variables

Here, $x \in \mathcal{X}$ is the observed data (e.g., an image); $\hat{z} = E_{\text{img}}(x)$ is the encoder feature before group action; $(\mu, \log \sigma^2) = E_{\text{group}}(\hat{z})$ are the parameters of the variational posterior; $t = \mu + \sigma \odot \varepsilon$ with $\varepsilon \sim \mathcal{N}(0, I)$ are Lie-algebra coordinates; $A(t) = \sum_j t_j A_j$ combines generator matrices $A_j$; $z = \exp(A(t)) \hat{z}$ is the Lie-transformed latent; and $D_{\text{img}}(z)$ denotes the decoder reconstruction.

## A.2   Proof of Theorem 1

*Proof.* Consider the $2 \times 2$ Jacobian block:

$$J = \left[ \frac{\partial x_{\text{rec}}}{\partial t_i} \,\middle|\, \frac{\partial x_{\text{rec}}}{\partial t_j} \right].$$

Using the chain rule, each column can be expressed as:

$$\frac{\partial x_{\text{rec}}}{\partial t_k} = \frac{\partial D_{\text{img}}}{\partial z} \frac{\partial z}{\partial t_k}, \quad k \in \{i, j\}.$$

Therefore, the Jacobian block factors as:

$$J = \frac{\partial D_{\text{img}}}{\partial z} \left[ \frac{\partial z}{\partial t_i} \,\middle|\, \frac{\partial z}{\partial t_j} \right] = \frac{\partial D_{\text{img}}}{\partial z} Z,$$

where $Z = \left[ \frac{\partial z}{\partial t_i} \,\middle|\, \frac{\partial z}{\partial t_j} \right]$ is the Jacobian of the group action.

The Gram matrix of $J$ is:

$$J^{\mathsf{T}} J = Z^{\mathsf{T}} \left( \frac{\partial D_{\text{img}}}{\partial z} \right)^{\mathsf{T}} \left( \frac{\partial D_{\text{img}}}{\partial z} \right) Z = Z^{\mathsf{T}} M(z) Z,$$

where $M(z) = \left( \frac{\partial D_{\text{img}}}{\partial z} \right)^{\mathsf{T}} \left( \frac{\partial D_{\text{img}}}{\partial z} \right)$ is the pullback metric induced by the decoder mapping, similar to the Riemannian geometric framework of [1]. The singular values of $J$ are the square roots of the eigenvalues of $J^{\mathsf{T}} J = Z^{\mathsf{T}} M(z) Z$. Therefore:

$$U_{ij}^{\text{manifold}} = \sigma_{\max}(J) = \sqrt{\lambda_{\max}(Z^{\mathsf{T}} M(z) Z)}.$$

This measures the maximum Riemannian distortion of the linear map from the $(t_i, t_j)$ parameter space to the data space. Our approach uses the deterministic pullback metric, unlike Arvanitidis et al. (2017), which adds variance Jacobians for Gaussian decoders. With uncertainty explicitly tied to commutator norms of latent generators, our approach isolates the algebraic contribution of non-commutativity.

$\square$

## A.3   Implementation: Efficient computation of Riemannian distortion

We operationalize Theorem 1 through an efficient implementation that avoids explicit Jacobian construction. The implementation directly computes the measure established in Theorem 1:

**Transformation construction**: The latent transformation is constructed as

$$z = \exp(M) \cdot \hat{z},$$

where the Lie algebra element is defined by

$$M = \sum_k t_k A_k,$$

with $\{A_k\}$ the generator basis and $\{t_k\}$ the corresponding coordinates. This construction preserves the manifold structure by exponentiating a linear combination of generators.

**Jacobian-vector products for Riemannian sensitivity**: The sensitivity is calculated with respect to the parameters $t_i$, flowing through the full transformation chain: $t_i \to M \to z \to x_{\text{rec}}$.

For each latent dimension $i$, we compute the direction of change in the transformed latent space $z$ caused by an infinitesimal change in $t_i$. Using a first-order approximation accurate for small transformations [12], this direction is:

$$V_i = \frac{\partial z}{\partial t_i} \approx A_i \cdot z.$$

This vector $V_i \in \mathbb{R}^{\dim(z)}$ points in the direction of maximal change in $z$ for a change in $t_i$.

We then compute the Jacobian-vector product (JVP) of the decoder $D_{\text{img}}$ at the point $z$ along the direction $V_i$:

$$\Delta x_i = \text{JVP}\big(z \mapsto D_{\text{img}}(z), \ z, \ V_i\big).$$

The vectors $\Delta x_i$ and $\Delta x_j$ are precisely the columns of the Jacobian block in Theorem 1:

$$\Delta x_i = \frac{\partial x_{\text{rec}}}{\partial t_i}, \quad \Delta x_j = \frac{\partial x_{\text{rec}}}{\partial t_j}.$$

**Spectral norm via power iterations**: We compute the joint sensitivity efficiently:

$$U_{ij}^{\text{manifold}} = \sigma_{\max}([\Delta x_i | \Delta x_j])$$

using power iterations on the $2 \times 2$ Gram matrix [9]:

$$G = \begin{bmatrix} \Delta x_i \cdot \Delta x_i & \Delta x_i \cdot \Delta x_j \\ \Delta x_j \cdot \Delta x_i & \Delta x_j \cdot \Delta x_j \end{bmatrix}$$

which is the Gram matrix $J^\intercal J = Z^\intercal M(z) Z$ from Theorem 1. This provides an efficient, numerically stable computation of the Riemannian distortion measure while avoiding explicit Jacobian construction.

## A.4 Local Jacobian Commutator Details

For a given input $x$, we obtain a latent point $z = \exp(A(t))\hat{z}$ with $\hat{z} = E_{\text{img}}(x)$ and $t \sim q(t \mid \hat{z})$. The decoder's Jacobian in the direction of generator $A_k$ is

$$J_k(z) = \frac{\partial}{\partial \epsilon} D_{\text{img}}\big(\exp(\epsilon A_k)\, z\big)\bigg|_{\epsilon=0}.$$

The decoder commutator $C_{ij}(z) = J_i(z)J_j(z) - J_j(z)J_i(z)$ is compared to the algebraic commutator direction:

$$v_{[ij]}(z) = \frac{\partial}{\partial \epsilon} D_{\text{img}}\big(\exp(\epsilon[A_i, A_j])z\big)\bigg|_{\epsilon=0}.$$

This helps to decide whether this response reflects true group structure or decoder artifacts. Cosine alignment $\mathcal{L}_{\text{align}}(i, j; z) = 1 - \cos\angle(C_{ij}(z), v_{[ij]}(z))$ quantifies decoder faithfulness: small values indicate proper algebraic alignment, large values indicate distortion.

## A.5 Fairness from the deformation-stability principle

Our framework provides a fairness criterion that applies directly to continuous latent spaces structured by Lie group generators. This definition follows from the deformation–stability principle, which requires that geometric uncertainty scales with algebraic non-commutativity. In the commutative case ($[A_i, A_j] = 0$), fairness means uncertainty should exhibit *additive consistency*: the total uncertainty in joint transformations should equal the sum of uncertainties from individual transformations. In the non-commutative case ($[A_i, A_j] \neq 0$), fairness requires that the total uncertainty exceed the sum of individual uncertainties, with the excess proportional to the commutator strength $\|[A_i, A_j]\|_F$.

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
