# OpenReview forum: "When Non-Commutativity Breeds Unfairness: A Geometric–Algebraic View of Uncertainty in VAEs"
_EurIPS.cc/2025/Workshop/UPLB — UPLB2025_

### Official Review · Reviewer_ekTC · 2025-10-17
**Assessment**

**Rating:** 5
**Confidence:** 2

**Review:**

The paper argues that VAEs systematically understate uncertainty when latent transformations are order-dependent, and it links the “amount of non-commutativity” directly to how much extra uncertainty a model should report. It introduces two a diagnostics to detect intrinsic non-commutativity from the latent generators and to spot spurious order effects from the decoder, and proposes ways to mitigate it.

Despite looking technically sound, the claimed relationship between UQ and non-commutativity is hard to understand from the paper for a reader who is not familiar with the methodology. It is not clear exactly how these results compare to standard UQ methods in ML, and the paper lacks an illustration or experiment that could help clarifying this.

---

### Decision · Program_Chairs · 2025-11-03

Accept (Poster)